# Technology and Quality of Life of Older People in Times of COVID: A Qualitative Study on Their Changed Digital Profile

**DOI:** 10.3390/ijerph191610459

**Published:** 2022-08-22

**Authors:** Alicia Murciano-Hueso, Antonio-Víctor Martín-García, Ana Paula Cardoso

**Affiliations:** 1Faculty of Education, University of Salamanca, 37008 Salamanca, Spain; 2CI&DEI, Instituto Politécnico de Viseu, 3504-510 Viseu, Portugal

**Keywords:** elderly, technology, digital profile, quality of life, COVID-19, semi-structured interview, Portugal

## Abstract

The situation caused by the COVID-19 pandemic brought negative consequences such as social isolation, limited access to routine health and social care services, and lack of self-esteem, especially for older people. In this context, technology took on an important role as the main means of communication and service delivery. The main objective of this study is to analyze the situation of the elderly and their access to technological resources in the time of COVID-19. Based on a qualitative methodology, 58 semi-structured interviews were conducted with people between 65 and 90 years old in Portugal. The results reveal specific difficulties in the use of this type of digital devices and a change in the digital use profile of this age group, characterized by more frequent use of digital devices such as smartphones, the incorporation of certain services such as video calls, and uses associated with communication and entertainment. This study shows that attitudes toward technology among the elderly should be studied further, and these results should be considered to develop and implement training programs specially designed for this age group in order to contribute to their well-being and quality of life.

## 1. Introduction

The speed with which people integrate digital technology into their daily lives is a differentiating element, key to understanding today’s society. The need to adapt to changes caused by this technological integration helps to understand aspects such as the human–machine relationship, certain interaction practices, personal relationships and the sense of belonging, access to information, and social integration of various social and age groups. A rise and diversification in social uses of technology is a *habitus tecnológico*, characterized by a particular type of possession, means of use, significance given to technology, and cultural, educational, and skills barriers, and, especially, a higher level of dependence on technology [1]. Some authors, for example [2] (p. 107), have used the term ‘technological ways of life’ to indicate a relationship with the world via interlinked technological systems, i.e., ways of life that are also increasingly more remote.

According to the Digital 2021 report, Internet users have risen by 316 million (7.3%) with 4.66 billion people worldwide; social network users are up 500 million to 4.2 billion. Above any other digital device, mobiles have become the ‘top’ screen with 5.22 billion users worldwide (66.6% of the total population). Global confinement caused by the COVID-19 pandemic has accelerated this digital presence even more. Social distancing and mobility restrictions have increased social isolation and accentuated loneliness, particularly in older people [3]. In this context, ICTs have stood out as one of the most important tools for emotional connection [4], favoring their expansion even more. The so-called ‘silver surfers’ have experienced the fastest growth on the main social platforms, with users aged over 50 showing the most growth compared to other age groups on Facebook and Snapchat (the Digital 2021 report, for example, indicated that users aged over 65 rose by approximately 25% over the last year on Facebook). More frequent uses of other technologies related to e-commerce and online purchases, medical care (telehealth and connected services to help with mental welfare), education, or remote working were also detected.

Focusing on the elderly, the result of this series of changes has highlighted at least two different realities. On one hand, there is a more active, healthy view of old age with more community participation [5,6,7]. On the other hand, the presence of digital technologies has emerged as a disturbing and conditioning element for the functional development of elderly people as they experience difficulties in their access and use [8,9], a fact that increases the generational digital gap [10,11], social inequality [12,13,14], and differences compared to other age groups [15].

Studying how the elderly use certain digital devices is thus leading many researchers to analyze processes of routinization of technology linked to aging. In other words, this refers to discovering a process with which digital processes become standard elements in the daily lives of age groups traditionally less prone to their use. For this reason, together with an interest in finding out how older people adopt, appropriate, and use certain digital devices, is the interest in facing and addressing the challenges they present to a social, cultural group with skills far removed from this type of device. Studies on the use and competence of the elderly regarding digital technologies have recently been channeled in the so-called life-based design (LBD) paradigm, which aims to discover how different technology devices are incorporated in daily contexts and how they help to improve quality of life [16]. To achieve this, experts agree on the need to implement sustainable social policies that focus on the aging process in their daily life environments (aging in place). Part of the reasoning behind this type of approach is based on classic works such as the ecological model of competence by [17] and particularly Lawton’s subsequent consideration [18] on the hypothesis of environmental proactivity. These ideas lead to considering the need to improve the digital competence of this age group as essential in order to face growing technological pressure and in contexts of environmental diversity. Thus, the social and educational goals are basically to favor social interaction, active participation, and social commitment among the elderly by using technological resources as a factor of successful, healthy aging [19].

Training in digital skills that favor social inclusion is therefore a priority and features in the social policies of the main international bodies. For example, one of the key objectives of the European Union Lifelong Learning Programme (LLP) is the promotion of active aging and intergenerational relations through learning and acquiring digital skills in formal, non-formal, and informal environments. Meanwhile, training for both the elderly and professionals and carers in basic competences, those which refer to activities of daily life (ADL), such as personal hygiene and diet, among other factors, and in instrumental activities of daily life (IADL), which include activities related to independent living (using a cash machine, transferring money online, using a mobile, etc.), are also socio-educational content to be promoted. Various technological projects thus offer websites and social blogs, social mentoring projects based on virtual platforms, social networks, etc., which seek to ensure truly active aging, optimizing opportunities for health, participation, and safety and, ultimately, improving quality of life as people age [6,20].

To justify these projects, we must first analyze a series of factors related to perception of technology and removing barriers to the elderly using digital technology. Current research into this subject attempts to find answers to questions such as how technologies are perceived and represented in the human mind, what role they play in their lives or in configuring digital identity, and how they are used based on certain socio-demographic or cultural factors. Barriers or factors that stand between the elderly and digital technologies are also sought [21,22,23].

In the first case, attitudinal, cognitive, and emotional components that make up certain mental representations which condition the use, or intention to use, a given technology have been analyzed. In this context, there has been a considerable rise in studies analyzing the use of technologies by users in recent years. The theoretical and empirical framework most widely used for this type of study are technology adoption models (TAM, TCP, UTAUT, etc.) [24,25,26,27]. As for the main barriers between the elderly and digital technologies, a variety of factors have been identified such as age, economic level, education and user experience, usability problems, etc. However, there is also another aspect related to distrust of technologies or certain risks related to privacy and intimacy [28]. These risks increase as total connectivity, ubiquity, and overexposure rise; for example, the loss of privacy when an older person follows a certain program designed specifically for cognitive (e.g., memory) or sensory training or rehabilitation. In these cases, rejection of these devices by the elderly is the same as for the use of a walking stick, hearing aid, or wheelchair. In other words, there is an effect of loss of privacy not desired by users as they feel that the existence of a possible specific functional need can be highlighted. Barriers such as anxiety using a computer, low perceived self-efficiency in the use of digital devices, and socio-cultural factors associated with fluid intelligence [28,29] have also been analyzed, as well as factors related to perceptual-sensory and/or motor deficits or perceived health, among other factors [16,30,31].

Most studies have been approached from the logic of quantitative research [32,33,34]. A qualitative approach can offer a more profound and direct complementary overview of this type of perception [34,35]. In this study, we present this type of approach, with the fundamental objective of analyzing the relationship between older adults and technological resources in the time of COVID-19 based on a qualitative research design.

## 2. Materials and Methods

### 2.1. Design

The main objective of the study is to analyze the situation and access of the elderly to technological resources in the time of COVID-19, attempting to discover factors related to perception of technology resources, impact on quality of life, and elimination of barriers to use in a small group of older adults when they cannot interact in person. The following specific objectives are proposed:Understand factors related to perception of technological resources and perceived impact on quality of life in using technology when they cannot interact in person.Delve deeper into intention to use technology in a healthcare or other emergency situation (e.g., COVID-19).

It is expected that:

**Hypothesis** **1** **(H1).**
*Older adults have incorporated technological resources and increased their use as a result of the pandemic.*


**Hypothesis** **2** **(H2).**
*Use of technology during COVID-19 lockdown has had a positive impact on their well-being and quality of life.*


**Hypothesis** **3** **(H3).**
*The situation caused by COVID-19 positively influences the intention to use technological resources by older adults.*


The study followed a qualitative, exploratory, descriptive, and contextual design, comprising in-depth semi-structured interviews. The flow diagram is shown in Figure 1.

### 2.2. Population and Sampling

Participants were asked to provide written voluntary informed consent after being given a detailed description of the study by research staff. One-on-one semi-structured interviews were then conducted by three experienced qualitative interviewers who were not previously known to the participants using a prepared interview guide to know their reality through their own personal experiences, incorporating their own subjectivities in the processes of obtaining and transmitting information by describing their personal itineraries [36].

Interview schedules were designed based on the social cognitive theory (Bandura’s theory) and more specifically on the technology acceptance model (TAM) [37], which included the exploration of their perception and uses of technological resources and intention to use. This decision is justified given that TAM is the most widespread general analysis framework at the international level to analyze various types of technological uses for all user profiles.

Data were stored and thematic analysis conducted using NVivo 12 (NVivo version 12 (Salamanca, Spain)) software, which facilitates the coding process through nodes. Inductive coding techniques were used. The process was conducted through 4 phases between February and November 2021: (1) contact with participants; (2) interview script development; (3) semi-structured interviews. All interviews were conducted face-to-face and recorded in audio format to facilitate subsequent transcription and analysis. Interviews had an average duration of 30/40 min; and (4) transcription and analysis of the interviews. NVivo 12 qualitative data analysis software was used according to a system of categories elaborated for two researchers that is shown in Figure 2 [37]. The credibility of the analysis was further enhanced by coding the data searching for themes by making use of two independent coders and any coding inconsistencies and discrepancies were discussed and agreed upon to keep codes in alignment.

According to the category system, of the total discourse, 1108 references were recorded referring to digital profile, perceptions, and intention to use technologies. Coding the information based on the pre-established category system, we worked with a discourse analysis focusing on 3 dimensions: changes in digital profile caused by COVID-19 pandemic, perceptions of technological resources in the time of COVID-19, and expectation of future use. The proposed dimensions and categories of analysis are included in Table 1. Defined on the basis of inductive processes, they are considered as the routes that facilitate understanding of the change in digital profile of the elderly and their perception of it.

## 3. Results

Results show the characteristics of the participants, and taking these into account for interpretation, the results of the qualitative analysis are presented by category level. The classification presented makes it possible to see the evolution of older adult digital profiles in the time of COVID-19 and analyze a series of factors related to technological perception, elimination of barriers to the use of digital technology by the elderly, and their expectation of future use.

### 3.1. Characteristics of the Participants

The sample participating in this study consisted of a total of 58 people from Portugal aged between 65 and 90 years (M = 72.58; SD = 2.38). Diversity was observed with respect to the profile of older adults based on the area of residence (67% rural area and 33% urban area), gender (66% women and 34% men), marital status (71% married; 21% widowed; 8% divorced), and educational level (10% no studies; 57% primary studies; 19% secondary studies; 14% higher studies).

### 3.2. Change in Digital Profile since the COVID-19 Pandemic

A digital profile analysis was conducted before and after the COVID-19 pandemic in accordance with the first objective. The results show a change in use and frequency of use of technological resources. The main results can be seen graphically in Figure 3.

#### 3.2.1. Changes in the Digital Profile of the Elderly

Results of the study show that before COVID-19, smartphones were the main digital devices used by the elderly to act and interact. They chose these devices to communicate with people who are far away, mainly their children and grandchildren, especially by using applications such as WhatsApp.


*“I have been using it to call my daughters for many years (they live in the United States) and we call them there”*
(Woman, 75 years old).


*“I mostly used the telephone to communicate with my daughter and granddaughters who lived abroad and less the computer”*
(Man, 76 years old).

The results of the interviews highlight that the elderly made sporadic use of technological resources before the COVID-19 pandemic. A lack of experience, digital skills, and patience are key determinants for not using them. The results also stress the importance of the lack of perceived use. Older people did not perceive this need to use them as there are other means of communication, they have other people around them to do things for them, or the perception that they are not suitable for their age group.


*“I never used to use any of that... only the landline. I knew that the internet and video calls existed, but I never tried it […] I didn’t really understand anything…”*
(Woman, 75 years old).


*“Digital technologies are essential for young people, it seems like their lives depend on them, but for older people like me, I thought that no using them would make no difference”*
(Man, 81 years old).

In order to discover whether the situation caused by the COVID-19 pandemic had influenced the digital profile of the elderly, the discourse focused on frequency of use of these technological resources and the incorporation of certain services as a result of the situation.

The results of this study show that the situation caused by the COVID-19 pandemic impacted the presence of the elderly in the virtual world. Smartphones are still ahead of any other digital device as the ‘top’ screen. However, a greater incorporation and frequency of use of technologies, especially those related to e-commerce and online purchases, medical care, entertainment, or communication, were detected. Video calls were the resource most frequently incorporated because of the COVID-19 pandemic.


*“I started having appointments with my doctor on the Internet and having video calls with my friends and grandchildren”*
(Woman, 68 years old).


*“With the situation caused by COVID-19, I started to use online payments […] Due to lockdown and my illnesses, I had to adapt and I found out how I could make payments, and I started to opt for online payments […] Now I use them whenever I need to”*
(Man, 69 years old).


*“I remember that, before the pandemic, my grandchildren wanted to show me and buy me a mobile to make video calls, I was a little stubborn and didn’t want to. Now I admit it was a mistake and I changed my opinion because we went a long time without seeing each other”*
(Man, 75 years old).


*“Now I use it more to stay in contact with the people I was with every day before this situation caused by COVID-19, and of course, also with other people I already had a relationship with”*
(Man, 79 years old).

#### 3.2.2. Motivations or Reasons for the Change in Digital Profile

For a more in-depth analysis of how the situation caused by the COVID-19 pandemic influenced the digital profile of the elderly, the discourse also focused on the decisive reasons behind the change in their digital profile. Results show that the main reasons for incorporating technological resources and increasing their frequency of use were mainly related to communication and entertainment. Staying informed and using certain services were also significant motivations.


*“I started using Facebook more because I spent more time at home and I want not in contact with anybody […]. I used this application to communicate with people and find out their news. It is useful because it is easy to access and makes life easier during the pandemic as it is easier and safer for me to communicate”*
(Woman, 66 years old).


*“I started using the mobile and making video calls as I was not with my family members […] I use it when in case of need, when I need something or have a problem and want to talk to my children and grandson […] I did not expect to use it if it weren’t for this pandemic”*
(Man, 66 years old).


*“I use this to stay more informed and to talk to my family”*
(Man, 67 years old).

These results thus reveal that the situation caused by the COVID-19 pandemic had impacted the digital profile of the elderly, further accelerating this digital revolution. Before this situation, the elderly used technological resources sporadically, especially as they were a means to stay in contact with their family at a time when families often lived far away. Lockdown led to greater incorporation and frequency of use of technologies, especially those related to e-commerce and online purchases, medical care, entertainment or communication. Although the smartphone was still the most widely-used device, there were many more uses and much more use, especially of services such as video calls. Therefore, the situation caused by COVID-19 was a key element for older people who already used technologies to use them even more. However, above all, it was a boost for those who practically did not use them to start. This entailed a greater perception of the benefits technology could provide.

The results specifically show their wish to be more communicative, entertained, and informed and to be able to access certain services they would otherwise be unable to access. Technologies were a key tool in preventing their exclusion. They could communicate with their environment; fighting the loneliness and sadness of being in lockdown, they were distracted by the entertainment provided, but, above all, they could be in contact with what was going on outside and have access to services (administrative, healthcare, others) that were impossible given the situation.

### 3.3. Benefits of Technologies: Perception of Use and Improved Quality of Life

Regarding the second objective, we analyzed how the elderly perceived technological resources during the time of COVID-19 for a more in-depth view of factors related to their perception, impact on their well-being, and quality of life and to the elimination of barriers to use for the elderly when they cannot interact in person. The main results can be seen graphically in Figure 4.

#### 3.3.1. Benefits of Technologies: Perception of Use and Improved Quality of Life

Results show that increased use of technological resources has had a positive impact on daily life. Greater use is favored by their positive perception. Perceived use of technological resources, especially those related to maintaining family and social relationships, has contributed to improving the social isolation and loneliness caused by social distancing and restricted mobility. Psychological manifestations present during this situation, such as depression, anxiety, or anguish, are minimized by technological resources. The results reveal that a significant emotional connection is established between the elderly and technological resources and that they play an important role in their lives as a key element in improving quality of life.


*“They’re useful, I think that the most positive aspects are easy communication with others and distraction”*
(Woman, 77 years old).


*“Since I use them I feel much happier, I can let off steam, talk about my day, know whether everything is okay with my children and granddaughters, talking with them makes my days better, if I didn’t have these technologies I would be a much unhappier person”*
(Woman, 73 years old).


*“Technological devices are very useful because they ended up helping to fight loneliness […] They distract you and give you the chance to communicate with your loved ones, so they give you comfort when you feel more alone”*
(Man, 78 years old).


*“It’s very useful and improves my life, a lot, I don’t feel alone and I have much less chance of catching the virus because I don’t have to leave the house”*
(Man, 81 years old).

#### 3.3.2. Perception of Problems or Barriers

The results of this study allow us to identify barriers or factors that come between the elderly and technological resources. Problems of usability as a result of a lack of experience in using them or low digital skills levels are the main barriers detected in this study. Older people mostly claim that they do not understand technological resources or know how to use them properly. Another aspect related to distrust of technologies or certain risks related to privacy and intimacy was also detected. Results reveal great concern for the dangers of their use. They also highlight the lack of available resources as one of the main barriers between the elderly and technology. The lack of connectivity caused by rural living is a key factor in the elderly not using these technological resources.


*“I think that technologies are very good, but they’re very difficult […] The positive side is that we can talk with the family at a distance, and the negative is that it is hard to use these new things […] What I found hardest and still find hard to get used to is always having my mobile one and knowing how to use it”*
(Woman, 68 years old).


*“I don’t understand anything about computers and new technologies, what I find hardest to understand is all the steps to be able to use new technologies”*
(Woman, 75 years old).


*“My problem was that the Internet was very slow [...] Nothing can be done as they told me it was because there was no network [...] I would need a better Internet connection”*
(Woman, 68 years old).


*“My opinion of digital technologies is positive, they are useful in daily life not just to contact our families as they keep us close to others, but also because they are important for business and trade as we can shop on the Internet, and they have been very important in health as doctors have given appointments on mobile phones […] For me, the use of these technologies has positive and negative aspects. Nowadays we don’t know how to live without a mobile phone as we can contact other whenever we need to and at any time, the negative part is that there are malicious people who use and abuse others with these technologies, so you have to be very careful”*
(Man, 71 years old).

### 3.4. Perception of the COVID-19 Situation as an Opportunity to Learn: Demands and Expectations of Use

Following the third objective, we analyzed how the elderly perceive the COVID-19 situation in the use of technologies. Results show that the COVID-19 pandemic has been an opportunity to use technological resources. The main results can be seen graphically in Figure 5.

The results reveal how the situation caused by COVID-19 meant that the elderly could not carry out the activities they normally did before the pandemic, especially during lockdown. Activities such as going to certain places or meeting with other people outside the household were not possible, causing a rapid growth in the incorporation of technological resources and greater frequency of use due mainly to the need for communication and contact with reality. The elderly feel that they started using many technological resources they had never even thought of before thanks to the lockdowns. Technologies began to play a major role in their lives, and they became aware of their benefits. Once lockdown came to an end, they clearly intended to continue using those technological resources they have incorporated because of this situation. The results show motivation to learn to use them and improve skills due to the great possibilities technologies could offer.


*“The pandemic has been an opportunity to use these devices because I’ve used them to communicate with people far away as we had no chance to see each other regularly and that is just what I needed as most of the people closest to us are far away”*
(Woman, 77 years old).


*“Yes, the pandemic was an opportunity to use these technological devices because if it weren’t for COVID-19 I wouldn’t know of some technologies that make a difference in my life. It was a great opportunity”*
(Woman, 73 years old).


*“Of course, I’ll keep using these technologies at my own pace […] Knowledge is always good and it takes up no space as it’s always useful […] The pandemic has shown that we must be up-to-date with these new technologies so that, even if we are far away, we are there and closer to each other”*
(Man, 80 years old).

As for the demands found, the results clearly show the need to have someone to help resolve problems. All the subjects interviewed said that when they are facing a problem related to the use of technologies, they ask for help from a family member or someone close by. The results highlight a clear demand for training. The main reasons given by interviewees who said they did not want training were mainly lack of patience or capacity, both associated with them feeling too old to learn.


*“I feel it’s very important that we receive training adapted to use […] I would love to learn more, it’s necessary”*
(Woman, 69 years old)


*“At my age, I’ve got no patience for training”*
(Woman, 72 years old)

Therefore, these results highlight that the elderly have a positive perception of technological resources. The situation caused by COVID-19 has been perceived as an opportunity to use them and become aware of their use in daily life. The main benefits found were related to communication, entertainment, being informed, and access to services. Technologies are an effective inclusion tool for this age group. Despite the self-perceived benefits, the elderly face several barriers to their use. Difficulty related to lack of experience or digital skills and lack of knowledge are key factors. Additionally, the lack of resources, such as an Internet connection, or self-perceived risks related to lack of privacy or possible theft were factors. However, despite these difficulties, the data reveal a great predisposition to continue using them primarily for their role as a key element in improving quality of life. The main demands lie in having help to use them and receiving training adapted to their circumstances and needs. This study allows us to identify barriers or factors that come between the elderly and technological resources.

## 4. Discussion

This study aims to analyze the relationship between the elderly and technological resources during the pandemic, comparing their use before and after the lockdown caused by COVID-19. As this is an age group with unique personal circumstances and which is especially vulnerable to the restrictions imposed due to global confinement, it is important to understand how the use of technological resources has evolved and how the elderly value these resources at various levels: from perception of use to expectation of future use, also encompassing the difficulties inherent in the complex use of technologies and associated emotional aspects.

The results obtained in this study have confirmed the three hypotheses linked to the two operational objectives set. Therefore, we can say that there has been a change in the digital profile of the elderly due to the COVID-19 pandemic, and this is primarily because of the need for communication and entertainment as well as access to certain financial and healthcare services that were otherwise impossible at that time. The perception of use of technological resources and their effect on quality of life are pivotal in the digital evolution of the elderly. Firstly, our findings show that before the pandemic, older adults already used smartphones as their top screen. Use was sporadic and the main reason was communicating with people who are far away, especially their children and grandchildren. They essentially exchanged messages using the WhatsApp application. These results are consistent with others found in the related literature [38,39], which show how technological advancements have offered remarkable opportunities to deliver care and maintain connections despite the need to stay physically separated.

Secondly, the use of technological resources has evolved because of lockdowns. The results show a greater frequency of use of these resources and the incorporation of certain services such as video calls. The main reasons behind these changes are communication and entertainment as well as staying informed and using certain healthcare and financial services [40]. Results corroborate prior studies that stress their desire to communicate and be entertained in order to reduce social isolation and loneliness [41,42] but also to be informed and access services that would otherwise not be possible [43,44]; they also indicate that perception of technological resources is positive [45]. There is a clear rise in perceived usefulness and a significant emotional connection has been established between the elderly and technological resources. In this sense, previous studies [46] point out that this connection is essential for the elderly in their process of adopting a given technology. They play a significant role in their lives as a key element in improving quality of life. Recent research highlights that technological resources contribute to improving self-perceived quality of life among the elderly [47], especially due to benefits related to improving healthy social relationships, emotional health, quality of time, or reduced levels of depression [48,49].

A third noteworthy aspect is that our results identify a series of barriers or factors standing between the elderly and digital technologies. Usability problems caused by misuse and low levels of digital skills; distrust of technologies and perception of possible risks; and a lack of technological resources are all evident. These results back up prior research that has identified barriers related to economic level, education, user experience, and usability problems, as well as anxiety or low perceived self-efficiency, or distrust and risk related to privacy and intimacy [50,51]. Barriers that lead to the cyber exclusion of this age group are especially important [52], so it is necessary to rely on mechanisms that minimize these barriers, such as digital literacy and initiatives such as Senior Living Lavs, which allow older adults to participate in a process of co-creation of devices and applications adapted to their needs [53].

Finally, the results highlight their clear intention to continue using technological resources. Some studies have stressed that older adults have increasing interest in using technologies to adapt and integrate in today’s society [54,55]. The results show that the COVID-19 pandemic has contributed to this rise as an opportunity to use technological resources. Consistent with prior research, this result seems to be logical as the need to start using certain technological resources has facilitated the group becoming familiar with them and being aware of their use, with greater motivation to use them for new opportunities [56].

## 5. Conclusions

The results of the study show an increase in the use of technological resources commonly used by the elderly and also the incorporation of new resources (particularly in terms of communication, entertainment, and the need to access financial and healthcare services) during the pandemic. In addition, it also contributed to a considerable increase in the perceived use of technological resources and a very positive perception of their use as a key element in the quality of life of older adults, probably because of the strong emotional linked between the elderly and technologies, particularly as they can break from the social isolation and loneliness inherent to confinement. They have expressed their clear intent to continue using the technological resources incorporated and to learn and develop digital skills.

Meanwhile, specific difficulties were also experienced in the use of these devices associated with factors such as: usability problems caused by misuse and low levels of digital skills; distrust of technologies and possible risks of use; and a lack of technological resources, especially insufficient connectivity in rural areas where a majority of the elderly live, a factor that is particularly noteworthy in Portugal.

Research such as this study therefore provides promising data that are worthy of note by political leaders, institutions, and families as they represent a unique opportunity to develop and implement training programs especially designed for this age group in order to contribute to the well-being and quality of life of the elderly.

## Figures and Tables

**Figure 1 ijerph-19-10459-f001:**
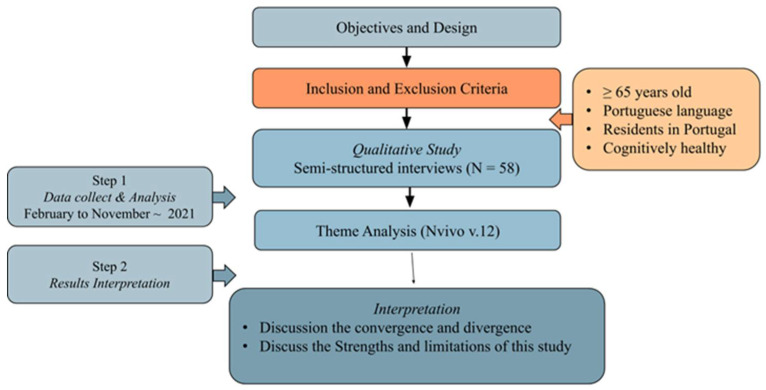
Flow diagram.

**Figure 2 ijerph-19-10459-f002:**
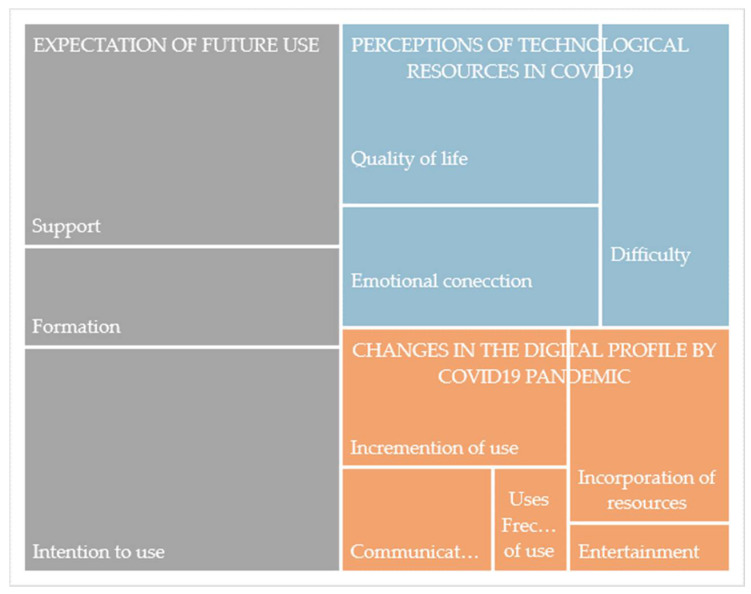
Hierarchical map of the discourse according to number of coding references. Words cut off by the graph: Communicat… = Communication; Frec… of use = Frecuency of use.

**Figure 3 ijerph-19-10459-f003:**
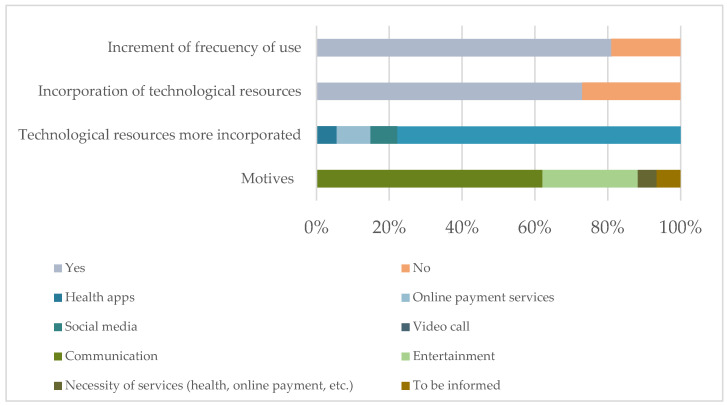
Main results of the digital profile of elderly as a result of COVID-19.

**Figure 4 ijerph-19-10459-f004:**
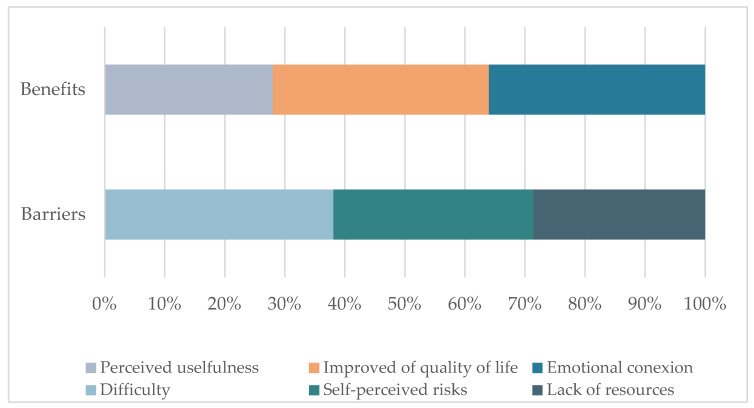
Main results of their perceptions of technological resources in the time of COVID-19.

**Figure 5 ijerph-19-10459-f005:**
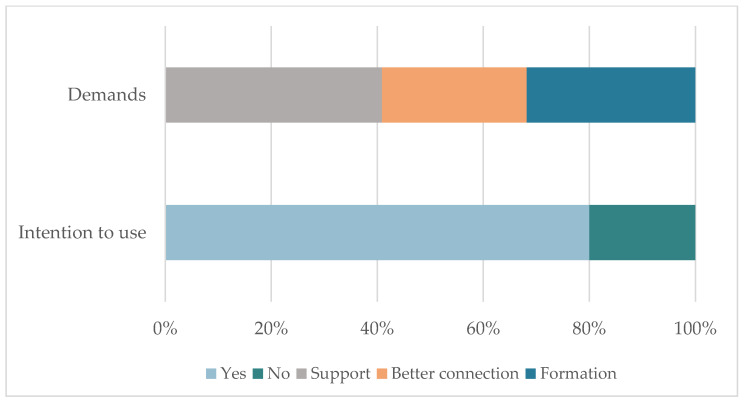
Main results for the expectation of their future use of technology.

**Table 1 ijerph-19-10459-t001:** Study dimensions and categories.

Dimensions	Categories	
Changes in digital profile caused by COVID-19 pandemic	Scenarios Pre-COVID-19	Uses
		Frequency of use
	Scenarios Post-COVID-19	Incorporation of resources
		Increased use
	Motivation for use	Communication
		Entertainment
		Information
		Services
Perceptions of technological resources in the time of COVID-19	Benefits	Usefulness
		Quality of life
		Emotional connection
	Barriers	Difficulty
		Self-perceived risks
		Lack of resources
		Low digital competences
Expectation of future use	Demands	Support
		Better connection
		Training
	Intention to use	

## Data Availability

Not applicable.

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
