# Peer review of "Technology and Quality of Life of Older People in Times of COVID: A Qualitative Study on Their Changed Digital Profile"

_ijerph, 2022, doi:10.3390/ijerph191610459_

Round 1

Reviewer 1 Report

Some minor changes are required. In page 1, it says for examen (this sounds as a mistake) and also in page 6 it says COVID-10...it must be COVID-19.

Why the authors only use TAM? Maybe it´s interesting that the authors consider another frameworks as UTAUT... Some more explanations is required for the decision made by choosing TAM as the theoretical framwork of analysis in this paper.

It would be interesting for this paper include some more references about grey divide and older adults ICT users (Quan-Hasse et al; Ragnedda; Van Dijk; Alonso et al)

Author Response

Dear revisor,

The authors are very grateful for the comments and suggestions for improvement made by the reviewer. The productive and valuable comments from the external reviewer allowed us to update many parts of the manuscript, as indicated in the responses to each of the comments below. In addition, all updated portions of the manuscript were highlighted in red so that the editor and reviewers could easily follow them.

  1. These changes have been made.
  2. The most important frameworks in the scientific literature on this topic are cited in the article. “The theoretical and empirical framework most widely used for this type of study are Technology Adoption Models (TAM, TCP, UTAUT, etc.) [2-274].” (P.3). However, following the reviewer's suggestions, the justification of the theoretical framework of analysis chosen in the article has been incorporated.

          “Interview schedules were designed based on the Social Cognitive Theory (Bandura’s Theory) and more specifically on the Technology Acceptance Model (TAM) [38] which included the exploration of their perception and uses of technological resources and intention to use. This decision is justified given that TAM, is the most widespread general analysis framework at the international level to analyse various types of technological uses for all user profiles. (P.4)”.

  3. A few more references to the grey divide and older ICT users have been incorporated.
  • Quan-Haase, A.; Williams, C.; Kicevski, M.; Elueze, I.; Wellman, B. Dividing the Grey Divide: Deconstructing Myths About Older Adults’ Online Activities, Skills, and Attitudes, The American Behavioral Scientist, 2018, 62(9): 1207–1228. https://doi.org/10.1177/0002764218777572  
  • Ragnedda, M. New Digital Inequalities. Algorithms Divide. In Enhancing Digital Equity, Cham, 2020 (pp. 61-83)

According to the results presented, we consider that we have followed all the indications for the article to be considered suitable for publication in your journal. However, if after a further review, the reviewer consider that we should continue working on it, their suggestions will be welcome, given our great interest in being able to contribute with our work to the editorial line and the publications accepted by your Journal.

Sincerely yours,

The authors.

Reviewer 2 Report

First of all, I would like to congratulate the authors for their work. It is a current and rigorous work. The qualitative methodology is necessary to know the benefits and the real barriers that prevent the older population from accessing and using technology.

Here are some points that need to be revised in your manuscript:

1. Introduction.

The introduction deals with very interesting topics, however it is very long and does not sufficiently reflect the scientific production on the subject and what their work contributes.

The authors claim that there is only quantitative work on the use of technology in times of covid and older people. I suggest that the authors conduct a literature review. They will find very interesting qualitative methodological work (Sunyoung Kim 2022; Anu-Marja Kaihlanen 2022; Elizabeth A. Albers 2022; Desai 2022….);

 2.        Results:

- Important data on the sample of subjects have been omitted. The interpretation of the results described depends on the characteristics of the participants: age, gender, level of functional independence, place of residence... These data should be included in the results section.

3.         Discussion:

The authors comment on their results and make few references to previous studies. For example, in the second paragraph of the discussion, the authors comment: "These results are consistent with others found in related literature [32,33]", without elaborating on why they are consistent with the results of previous works.  This "tangential" way of commenting on previous research is continually repeated in the discussion.

On the other hand, the authors have found barriers in the use of technology, although this was not part of the hypotheses. In the discussion, how to minimise these barriers should be addressed.

4.      The conclusion is too long. It should be reduced to a maximum of three paragraphs. 

Author Response

Dear reviewer,

The authors are very grateful for the comments and suggestions for improvement made by the reviewer. The productive and valuable comments from the external reviewes allowed us to update many parts of the manuscript, as indicated in the responses to each of the comments below. In addition, all updated portions of the manuscript were highlighted in red so that the editor could easily follow them.

  1. In the introduction we tried to delimit the research problem of our study, presenting how specialised research is addressing the factors related to perception of technology and removing barriers to the elderly using digital technology. The topic itself is very broad, novel and still to be explored, so we believe it is of interest to extend the presentation a little bit. However, according to the suggestions, quantitative and qualitative studies have been incorporated to enrich this section.

    • Albers, E. A.; Mikal, J.; Millenbah, A.; Finlay, J.; Jutkowitz, E., Mitchell, L.; ... Gaugler, J. E. The Use of Technology Among Persons With Memory Concerns and Their Caregivers in the United States During the COVID-19 Pandemic: Qualitative Study. JMIR aging, 2022, 5(1), e31552. https://doi.org/10.2196/31552

    • Kaihlanen, A. M.; Virtanen, L.; Buchert, U.; Safarov, N.; Valkonen, P.; Hietapakka, L., ... ; Heponiemi, T. Towards digital health equity-a qualitative study of the challenges experienced by vulnerable groups in using digital health services in the COVID-19 era. BMC health services research, 2022, 22(1), 1-12. https://doi.org/10.1186/s12913-022-07584-
    • Murciano-Hueso, A.; Martín-Lucas, J.; Serrate-González, S.; Torrijos Fincias, F. (2022). Use and perception of gerontechnology: differences in a group of Spanish older adults. Quality in Ageing and Older Adults,2022, 23(3) [In press].

2. Data on the sample of participants have been included in the results section.

3. The format of the discussions has been modified and references relevant to the research have been incorporated. On the other hand, anumber of mechanisms have been incorporated in the article to minimise these barriers.

4. The conclusions section has been summarised.

According to the results presented, we consider that we have followed all the indications for the article to be considered suitable for publication in the journal. However, if after a further review, the reviewer consider that we should continue working on it, their suggestions will be welcome, given our great interest in being able to contribute with our work to the editorial line and the publications accepted by the Journal.

Sincerely yours,

The authors.

Round 2

Reviewer 2 Report

Dear authors,

I appreciate the changes made to the manuscript. I believe it is now more complete.

Regards,